# Effect of Antimicrobial Treatment on the Dynamics of Ceftiofur Resistance in Enterobacteriaceae from Adult California Dairy Cows

**DOI:** 10.3390/microorganisms9040828

**Published:** 2021-04-14

**Authors:** David B. Sheedy, Emmanuel Okello, Deniece R. Williams, Katie Precht, Elisa Cella, Terry W. Lehenbauer, Sharif S. Aly

**Affiliations:** 1Veterinary Medicine Teaching and Research Center, School of Veterinary Medicine, University of California, Davis, Tulare, CA 93274, USA; dbsheedy@ucdavis.edu (D.B.S.); eokello@ucdavis.edu (E.O.); dvmwilliams@ucdavis.edu (D.R.W.); kmprecht@ucdavis.edu (K.P.); ecella@ucdavis.edu (E.C.); tlehenbauer@vmtrc.ucdavis.edu (T.W.L.); 2Department of Population Health & Reproduction, School of Veterinary Medicine, University of California, Davis, Tulare, CA 95616, USA

**Keywords:** antimicrobial resistance, ceftiofur, medically important antimicrobial drugs, dairy, minimum inhibitory concentration

## Abstract

Dairy farm use of antimicrobial drugs (AMD) is a risk for the selection of antimicrobial resistance (AMR); however, these resistance dynamics are not fully understood. A cohort study on two dairy farms enrolled 96 cows with their fecal samples collected three times weekly, for the first 60 days in milk. Enterobacteriaceae were enumerated by spiral plating samples onto MacConkey agar impregnated with 0, 1, 8, 16 and 30 µg/mL ceftiofur. Negative binomial regression analyzed AMR over time. The continuum of ceftiofur concentrations permitted estimation of the minimum inhibitory concentration (MIC) and analysis using interval regression. The most common systemic AMD was ceftiofur, administered in 94% of treatments (15/16 cows). Enterobacteriaceae did not grow in 88% of samples collected from non-AMD treated cows at 8 µg/mL ceftiofur. Samples from AMD treated cows had peak counts of resistant Enterobacteriaceae during AMD treatment and returned to baseline counts by 3–4 days post-treatment at 8 µg/mL. Sensitive Enterobacteriaceae (0–1 µg/mL ceftiofur) were reduced below pre-treated levels for 29–35 days post-AMD treatment. Population MIC peaked during AMD treatment and returned to baseline levels by 7–8 days. We conclude that the effect of systemic ceftiofur on the resistance of Enterobacteriaceae in early lactation dairy cows was limited in duration.

## 1. Introduction

Antimicrobial resistance (AMR) in bacterial infections continues to rise globally and is reported to be responsible for approximately 82,000 deaths annually in the European Union and the United States [1,2]. The One Health concept and antimicrobial stewardship guidelines necessitate that the food animal industry investigate on-farm antimicrobial drug (AMD) use as a risk factor for the selection and emergence of antimicrobial resistant bacteria [3], hereafter referred to as AMR bacteria. Stewardship practices for on-farm AMD use have been promoted with guidelines and regulations to reduce and control AMR [4,5,6]. Particular focus is targeted towards the list of Medically Important Antimicrobial Drugs (MIADs), compiled by the World Health Organization, defined as AMDs used in both human and veterinary medicine [7]. Furthermore, MIADs are ranked on importance and priority based on a set of criteria that includes whether the antimicrobial class is the sole, or one of limited available therapies, to treat serious bacterial infections in humans [7].

Ceftiofur is a broad-spectrum 3rd generation cephalosporin, a class of AMD that belongs to the MIAD rank of “Highest Priority, Critically Important” [8], primarily for its chemical similarity to cephalosporins that are administered as one of few available therapies to treat Salmonellosis and *Escherichia coli* (*E. coli*) infections in humans [4,9]. Ceftiofur is also a frequently used AMD treatment option in adult lactating dairy cows, within the United States, for disease complexes including bovine respiratory disease (78% of treated cases), lameness (59%), diarrhea/digestive disorders (57%), metritis (46%) and mastitis (34%) [10,11,12]. In addition to its broad-spectrum therapeutic range, ceftiofur is an attractive AMD choice for producers as parenterally administrated formulations carry a zero-hour milk and short pre-slaughter withdrawal periods. A study on cull dairy cows identified 10% of *Salmonella* isolates as resistant to ceftiofur [13], whilst 1.9% of *E.coli* were resistant (at ≥8 µg/mL) in healthy dairy cows [14].

Enteric *E.coli* is commonly used as an indicator organism for Gram-negative bacteria resistance to AMDs, including ceftiofur, due to its predominance (~87%) in culturable Gram-negative enteric bacteria and ability to provide information on the reservoir of AMR genes that may be transferred, through horizontal gene transfer (HGT), to other pathogenic bacteria [15,16]. Under AMD selective pressure, *E.coli* readily acquires AMR through HGT, often via direct conjugation with other AMR gene bearing bacteria [17,18].

Bacteria that possess AMR genes may incur a biological fitness cost, limiting their survival compared to non-AMR bacteria [19]. The dynamic relationship between ceftiofur resistant and non-resistant populations of fecal *E.coli* has previously been explored in dairy and beef cows through longitudinal studies that determined AMR of select individual colonies using the broth microdilution [20,21,22] or agar diffusion methods [23] or by detecting the presence of the beta-lactamase encoding gene (CMY-2) or other known AMR genes [21,22,23]. An alternative method, that had not been explored in dairy cows using frequent follow-up intervals (2–3 days), is to compare fecal *E.coli* concentrations in the presence or absence of known ceftiofur concentrations [24,25]. Such a method increases the accuracy of estimates for phenotypically ceftiofur-resistant bacteria, compared to broth microdilution or agar diffusion methods, by observing a much larger number of bacteria per sample, and when performed with high sampling frequency can provide new insights into the dynamics of AMR bacteria.

The objective of this prospective cohort study was to investigate the time to peak population and time to return to pre-treatment population of phenotypically ceftiofur-resistant enteric commensal bacteria in response to systemic antimicrobial treatment, in commercial dairy cows during the first 60 days of lactation, on two California dairies each across two seasons.

## 2. Materials and Methods

### 2.1. Study Population

A free-stall Holstein dairy herd in the San Joaquin Valley of California (Farm A), milking 800 cows, was enrolled by convenience sample and willingness to participate in the study. The initial protocol was to collect data from two cohorts of cows followed from pre-calving up to 120 days in milk (DIM), across two distinct time periods. A decision to alter this design was made after 30 days of sample collection due to low treatment frequency with systemic antimicrobials (0% of enrolled cows) on Farm A. Therefore, a second San Joaquin farm (Farm B) was subsequently recruited, and cohort sampling duration was reduced to 60 DIM for both farms. Farm B was a Jersey, dry-lot dairy herd milking 2650 cows. Farm A had off-site calf rearing while Farm B raised calves on-site. The first cohorts were sampled during the Fall/Winter season (Farm A: 15 October–30 December 2018, Farm B: 10 December 2018–20 February 2019) and the second cohorts were sampled during Spring/Summer season (Farm A: 4 March–27 May 2019, Farm B: 20 May–5 August 2019).

### 2.2. Enrollment

Eligible cows had to be pregnant and have an expected calving date within 4 weeks from enrollment. Ineligible cows were those likely to be sold during early lactation (according to the farm manager) or with obvious clinical illness determined by the enrolling veterinarian. A list of eligible cows was generated using each farm’s dairy management software (Dairycomp305, Valley Ag software, Tulare, CA, USA), from which 24 cows per farm per cohort were randomly selected (Excel, Microsoft Corp, Redmond, WA, USA), controlling for parity to match the lactating herd structure: categorized as 1st, 2nd and 3+ parity groups. Selected cows were visually marked by spraying tail-paint transversely across their tuber coxae and the paint was re-applied as necessary during the follow up period. Farm managers/workers and on-farm research personnel were not blinded to selected cows, although study personnel performing sample processing and antimicrobial resistance testing procedures were blinded. Participating dairies and enrolled cow data were de-identified and stored on a relational database that enforced referential integrity (Access, Microsoft Corp, Redmond, WA, USA). Animal use was approved by UC Davis Institutional Animal Care and Use Committee (Protocol number 19871; approved 22-May-2018).

### 2.3. Sample Collection

Figure 1 depicts the process from sample collection through to the enumeration of Enterobacteriaceae. Fecal samples were collected at enrollment (pre-calving) and three times a week onwards (every Monday, Wednesday and Friday) post-parturition until 60 DIM or until the cow was sold or died. Approximately 40 g of individual fecal samples were manually collected from the rectum of enrolled cows with disposable polyethylene sleeves, changed between each cow, with a small amount (<3 mL) of sterile lubricant applied. Feces were placed in individual, sterile, 50 mL polypropylene tubes, transported to the Veterinary Medicine Teaching and Research Center (VMTRC) on wet ice, and refrigerated at 4 °C until processing.

Individual cow fecal scores were recorded at each visit using a 3-point scale; 1 = normal feces, 2 = loose feces and 3 = watery feces. Body condition scores (1–5 scale) [26] were recorded at enrollment and monthly thereafter. Any health events, treatments, pen changes or otherwise notable events were stored on both paper and electronic records. DairyComp305 herd record backups were collected weekly.

### 2.4. Sample Processing

Fecal samples were processed for long-term storage within 18 h of collection. Feces were homogenized using a sterile wooden spatula via 10 vertical stirring motions followed by 5 clockwise and 5 counterclockwise stirring rotations. Using a scale, 7 g of each fecal sample were weighed and mixed with 7 mL of 0.1 M Tris(hydroxymethyl)aminomethane (TRIS) buffer and vortexed for 5 s in a 50 mL polypropylene tube, to further homogenize the sample. Performed in quadruplicate, 1 g of the fecal/TRIS mixture was added to 600 μg of Trypticase Soy Broth (TSB) with 70% glycerol in 2 mL microcentrifuge tubes and vortexed for 5 s. The processed samples were stored at −80 °C until enumeration.

### 2.5. Ceftiofur-Impregnated MacConkey Agar

Bacterial enumeration was performed on MacConkey agar (a selective growth media for Gram-negative and lactose-fermenting bacteria) impregnated with ceftiofur. Though the enumerated commensal colonies were assumed to be *E. coli*, it was impractical to differentiate individual colonies beyond their phenotypic presentation and therefore the family Enterobacteriaceae is reported. Media plates were prepared by mixing 25 g of Difco^TM^ MacConkey Agar (Becton, Dickinson and Company, Franklin Lakes, NJ, USA) in 500 mL of de-ionized water and autoclaved at 15 PSI, 121 °C for 15 min. A titrated solution of ceftiofur (Thermo Fischer Scientific, Ward Hill, MA, USA) was added to the autoclaved media after cooling to 50 to 55 °C to make final concentrations of 0, 1, 8, 16 and 30 μg ceftiofur/mL, poured into standard 100 mm × 15 mm Petri dishes, and stored at 4 °C. During preliminary validation of the lab protocols, 0.3, 1, 3, 4, 8, 16 and 30 µg/mL of ceftiofur were investigated. The inclusion of 1 µg/mL was made as the lowest concentration that showed consistently less growth than the no-ceftiofur media. The concentration of 8 µg/mL was chosen as a reported clinical resistant cut-off point [27]. In addition, 30 µg/mL was chosen to identify Enterobacteriaceae resistant at the concentration used for a similar study to determine rates of gain and loss of resistance to ceftiofur in calves [28]. Furthermore, 16 µg/mL was added as a midpoint between the CLSI reference cut-off for ceftiofur resistance and 30 µg/mL.

### 2.6. Bacterial Enumeration

The stored fecal samples in TSB/25% glycerol media were thawed and resuspended in 3.4 mL of sterile 0.1 M TRIS buffer, to constitute a 1:10 final dilution of the fecal samples. A Whitley Automated Spiral Plater 2 (WASP 2, Microbiology International, Frederick, MD, USA) was used, according to the manufacturer instructions, to spread 50 μL aliquots (logarithmic mode) of the diluted fecal sample onto MacConkey agar plates. Each sample was spread in duplicates onto 0, 1, 8, 16 and 30 μg ceftiofur/mL concentration plates. The plates were incubated for 24 h at 37 °C and Enterobacteriaceae enumeration was performed using the SphereFlash Automatic Colony Counter (IULmicro, Barcelona), counting only the lactose-fermenting “pink colonies” and with manual edits to correct misclassified colonies or artefacts, as necessary. Further characterization of colonies was not performed. Results were recorded as CFU/g of feces. Photographs of enumerated plates were digitally stored. In circumstances where accurate enumeration was not possible due to high colony counts (~10^7^ CFU/mL, >400 colonies in the outside spiral) two additional 10-fold serial dilutions were performed on the fecal solution, re-plated and the lowest dilution capable of accurate bacteria counts enumerated and recorded for analysis.

Farm B was enumerated first, as we hypothesized this farm would have the greatest prevalence of resistant bacteria due to treatment frequency and could provide feedback to alter the above protocol as necessary. Preliminary results from the first cohort of Farm B were evaluated before commencing the enumeration of the other cohorts. The counts of resistant bacteria grown on the 16 μg/mL ceftiofur-impregnated MacConkey plates closely matched the results from the 8 μg/mL ceftiofur plates. Subsequently we decided to not use the 16 μg/mL plates for Farm A enumeration to optimize cost and time efficiency with little expected impact on analytical power.

### 2.7. Positive/Negative Controls

A ceftiofur-susceptible *E.coli* strain (*Escherichia coli* ATCC 25922^TM^, Manassas, VA, USA) was used for negative control, and positive control strains were comprised of two ceftiofur-resistant *E. coli* field strains (isolated at the VMTRC from a dairy calf, and from an adult Farm B cow; both resistant at ceftiofur ≥30 µg/mL). Control strains used in the experiments were freshly picked from the −80 °C stock (in TSB/10% Glycerol) every two weeks and streaked onto sheep’s blood agar (SBA). Each day spiral plating was performed, a control strain colony on SBA was cultured into 10mL of TSB for 4 to 12 h at room temperature. An aliquot of the cell culture was diluted 1:4 in phosphate buffered saline and had its cell concentration estimated by measuring its optical density at 600 nm (OD_600_). This value was used to calculate the required dilution to create a suspension at 10^6^ and 10^4^ CFU/mL. These cell suspensions were spiral plated as per the above protocol.

### 2.8. Statistical Analyses

For all analyses, the bacterial count data for each ceftiofur concentration/cow/sample day was calculated as the mean of the two replicates. Statistical significance was considered at *p*-values less than 0.05. All the statistical analyses were conducted using Stata IC 16 (College Station, TA, USA). To estimate Enterobacteriaceae counts that were sensitive between 0 and 1 µg/mL of ceftiofur, the 1µg/mL ceftiofur counts were subtracted from the 0 µg/mL counts for each cow-day. Herein this estimate is referred to as sensitive Enterobacteriaceae. For the 14 instances (0.62%) where the 1 µg/mL ceftiofur counts were greater than the 0 µg/mL counts, sensitive Enterobacteriaceae was estimated as 0 CFU/g.

### 2.9. Kernel-Weighted Local Polynomial Smoothed Regression

A graphical analysis of the observed Enterobacteriaceae counts (CFU/g) for treated and untreated cows over time was performed using kernel-weighted local polynomial smoothing regression, without requiring assumptions for the underlying data distribution. The *x*-axis was days post-antibiotic treatment (*dpat*), with *dpat* = 0 being the last day of AMD treatment and the *y*-axis was Enterobacteriaceae counts (CFU/g). For each treated cow, there were five randomly selected untreated cows that were assigned *dpat* values such that their DIM and *dpat* were identical to that of their matched treated cow, to control for different DIM at treatment completion amongst cows. As duration of treatment also varied amongst cows (range: 3–12 days), negative *dpat* values include pre-treated and treated cows and hence should be interpreted as such. Figures for the four ceftiofur concentrations and the sensitive estimate were produced.

### 2.10. Multilevel Mixed Effects Negative Binomial Models (MENBM)

Enterobacteriaceae counts (CFU/g) for each concentration of ceftiofur-impregnated plates and the sensitive count estimate were specified as dependent variables in MENBMs, excluding the 30 µg/mL ceftiofur counts due to the failure of model convergence. The over-dispersed count data was not suitable for a Poisson model (which assumes that the mean = variance; for ceftiofur 0 µg/mL plates, mean = 1.99 × 10^5^ and variance = 1.37 × 10^11^) and was formally tested by comparing the log-likelihoods of the univariate Poisson model and the MENBM. The equation below describes the full MENBM equation predicting $\mu_{ij}$, mean count of Enterobacteriaceae: [29]
$$\mu_{ij}=e^{\left(\begin{array}{c} \beta_{0}+\beta_{1}Season_{i}+\beta_{2}Farm_{i}+\beta_{3}Parity_{i}+\beta_{4}Fecal\;Score_{ij}+\beta_{5}Mastitis_{ij}\\ +\beta_{6}DPT_{ij}+a_{0i}^{cow}+b_{1i}^{Cow}DIM_{ij}\; \end{array}\right)}$$

The MENBM predicts $\mu_{ij}$, the conditional expectation of the observed Enterobacteriaceae counts ($y_{ij}$), in fecal samples collected from the *i^th^* cow at its *j^th^* DIM (ranging from −24 to 63, where negative DIM represents the pre-calving sampling event).

The observed $y_{ij}$ counts are distributed conditional on the random intercept $a_{0i}^{cow}$ for the *i*th cow and a random slope coefficient $b_{1i}^{Cow}$ at the cow level for DIM, with a Negative Binomial distribution with number of successes *r*, and probability of success $p_{ij}$, a special case of Generalized Linear Mixed Models. The model random effects *a_i_* (*i* = 1, 2, …, m) and *b_ij_* (*j* = 1, 2, …, n_m_) are independent and distributed as N(0; $\sigma_{a}^{2}$) and N(0; $\sigma_{b}^{2}$), respectively. Including farm as a higher first-level random intercept was not possible due to poor model convergence; instead robust standard error estimates were calculated with a clustered sandwich variance estimator that allowed for intragroup (farm-level) correlations.

The model fixed effects included the intercept ($\beta_{0}$) and categorical variables for season (Fall/Winter, reference; and Spring/Summer), dairy (Farm A, reference; and Farm B), parity (1, reference; 2; and ≥3), fecal score (1, reference; 2; and 3) and a variable for the AMD treatment follow up timeline (Days Post-Treatment: DPT) was forced into the model, given the study objective. To specify DPT with respect to when systemic AMDs were administered, the variable’s reference level represented Enterobacteriaceae counts in samples collected from all untreated cows and prior to treatment in treated cows; a dummy variable representing samples collected during treatment; and additional dummy variables representing samples collected post-treatment. Different specifications for the post-treatment dummy variables were compared including 1, 2, 3, 4, 5, 6 and 7-day interval periods. Furthermore, the periods of time post-treatment with no significant differences compared to pre-treatment were grouped by either 1- or 2-week intervals and model fit assessed. An identical approach for specifying a time variable for intramammary AMDs was also performed. A manual model building approach was adopted while exploring confounding using the method of change in estimates and all two-way interactions using statistical testing at the 5% level of significance [30]. The best fitting model was explored by comparing observed versus predicted plots of Enterobacteriaceae counts and confirmed using the Akaike Information Criterion (AIC), where a lower AIC estimate resembled a better fit.

### 2.11. Multilevel Mixed Effects Interval Regression Models

To model the effect of treatment on fecal MIC over time, two types of mixed effect interval regression models were created. Interval regression models fit continuous responses where the dependent variable is measured as interval censored data. The interval regression models estimated the MIC in treated cows utilizing growth of Enterobacteriaceae at the ceftiofur concentration (µg/mL) intervals of: [0, 1], [1, 8], [8, 16], [16, 30] and [30, +∞). The difference between the two types of MIC interval regression models was in determining their interval censored data, with Table 1 showing numerical examples of two methods. In the “Maximum MIC” model, for each cow-sample-day there was one interval assigned; the highest concentration of ceftiofur MacConkey agar plate that permitted at least a single colony to grow was assigned the interval lower endpoint. The “Population MIC” model allowed multiple intervals per cow per sample; estimated colony counts (CFU/g) for each interval were calculated as IntervalCount[*n*, *n_+_*_1_] = ColonyCount[*n*] − ColonyCount[*n_+_*_1_], where *n* = the interval lower endpoint ceftiofur concentration and *n_+_*_1_ = the interval upper endpoint ceftiofur concentration, with the exception of the right censored interval [30, *+∞*), which was the unaltered colony count for the 30 μg/mL ceftiofur plate (Table 1). There were 92 (5.5%) instances where the *n_+_*_1_ concentration of ceftiofur had more growth than the *n* concentration (median difference 300 colonies, SE = 16,778). In these circumstances the IntervalCount [*n*, *n_+_*_1_] was increased to zero, to avoid negative interval counts.

The random effect structure specified for the mixed effects interval regression models included a random intercept for the individual cows and DIM as a random covariate (slope). Only Farm B cows treated with systemic antimicrobials were included in these models. The variable DPT, as specified in the MENBM, was forced into the interval regression models given the study objective. Season, DIM, lactation number, fecal score and mastitis treatment were investigated as fixed effects.

## 3. Results

### 3.1. Study Population and AMD Treatments

A total of 96 of 259 eligible cows were enrolled in the study (see Table 2), with 5 excluded from analysis due to uncertain treatment records (*n* = 2) and no post-parturient samples (*n* = 3, due to being sold before a second sample was taken). Farm A increased the proportion of first lactation heifers from 25% to 50% between seasonal cohorts whilst Farm B maintained a stable herd structure with regard to parity. The mean length of follow-up period that cows were sampled for was 55.7 DIM (SE = 1.47) with 9 cows sold (Farm A; *n* = 1, Farm B; *n* = 8, 3 treated with an AMD) and 4 cows that died (Farm A; *n* = 2, recorded causes: diarrhea, unknown, Farm B; *n* = 2, recorded causes: diarrhea—treated with penicillin, metritis—treated with ceftiofur) prior to completing the intended 60-day sampling period.

Farm A did not treat any of the enrolled cattle with systemic AMD. Farm B treated 36% (16/44) of the enrolled cows, with the first lactation heifers treated more commonly (56%, 9/16) than cows of greater lactation (22%, 7/32). The most common systemic AMD was ceftiofur hydrochloride (Excenel, Zoetis, Parsipanny, NJ, USA) used in 94% (15/16) of treatments, with penicillin (PenOne Pro, VetOne, Boise, ID, USA) the only other systemic AMD, used in 12.5% (2/16). One cow was treated consecutively with ceftiofur then penicillin during a single illness event. Observation of the penicillin-only treated cows’ count data showed strong co-selection for ceftiofur resistance, and hence this cow’s data was analyzed as an AMD-treated cow in our analysis. The most commonly stated reason for treatment was fever (*n* = 7), metritis (*n* = 5), metritis with a fever (*n* = 3) and scours (*n* = 1). No cow had more than one illness (excluding mastitis) recorded during the 60-day sampling period. The most common treatment duration was 3 days (63.5%, 10/16) with a mean duration of 4.9 days (SE = 0.742, range [3,4,5,6,7,8,9,10,11,12]). Intramammary AMD treatments were given to 3 cows at Farm A and 1 cow in Farm B, and the products used were Pirsue (pirlimycin hydrochloride, Zoetis) and Spectramast LC (ceftiofur hydrochloride, Zoetis). Herd-wide treatment levels in early lactation cows (<60 DIM), during the study period, for Farm A was approximately 4.0% for both systemic and intramammary AMDs, respectively. Farm B treatment levels were 38.8% and 23.8% with systemic and intramammary AMDs, respectively.

The recorded proportion of zero count CFU/g for ceftiofur-resistant Enterobacteriaceae in untreated cow was 78.4% (1489/1900 cow sample days), 87.8% (1670/1903), 89.0% (634/712) and 95.6% (1857/1903) for 1, 8, 16 and 30 µg/mL of ceftiofur, respectively. In contrast, the proportion of zero count CFU/g for cows that were currently receiving treatment was 23.3% (7/30 cow sample days), 30.0% (9/30), 33.3% (10/30) and 76.6% (23/30) for 1, 8, 16 and 30 µg/mL of ceftiofur, respectively.

### 3.2. Kernel-Weighted Local Polynomial Smoothed Regression

Local polynomial smoothed graphs for each ceftiofur concentration visually depicted changes in the observed Enterobacteriaceae counts (CFU/g) post-systemic AMD treatment compared to DIM time-matched non-treated cows (Figure 2). One additional treated cow was excluded from this analysis as there were no post-treatment samples taken (cow died during treatment). Figure 3 overlays the regressions for the treated cows only, at all ceftiofur concentrations. The emergence of ceftiofur-resistant phenotypes increased in prevalence during a treatment course and returned to pre-treatment levels within 10 days after treatment completion. A cyclical re-emergence of resistance, with ever-decreasing peak size, was observed at around 18 *dpat*, 30 *dpat*, 40 *dpat* and 55 *dpat* for ≥1 µg/mL ceftiofur resistance. This phenomenon is also observed at higher ceftiofur concentrations but at reduced magnitude. Sensitive bacteria displayed the converse response to treatment, with counts rapidly dropping with treatment before steadily increasing after treatment was completed, dropping again from 18 to 25 *dpat* and returning to pre-treatment counts by day 40. Both treated and their matched untreated cows were observed to decrease in sensitive bacteria counts after 40 *dpat*, until study completion at approximately 60 DIM.

### 3.3. Multilevel Mixed Effects Negative Binomial Models

The final form of the treatment variable was with dummy variables encoding pre-treatment, currently undergoing treatment (beginning one day after the first injection) and days post-treatment in two-day intervals until 14 days and 7-day intervals afterwards until study completion. The change from two-day to 7-day intervals was based on the loss of statistical significance at 14 days post-treatment in the 1 µg/mL ceftiofur-resistant bacteria counts. Duration of treatment and its interaction with the treatment dummy variable were not significant. Fixed effects of season, DIM, lactation number, fecal score and mastitis treatment were non-significant. Farm B had significantly greater counts in the sensitive Enterobacteriaceae and the ≥1 µg/mL ceftiofur resistant models. Farm A was removed from analysis in the ≥8 µg/mL to aid model convergence and was not a co-efficient for the 16 µg/mL as only Farm B had enumeration performed at this ceftiofur concentration.

The estimated coefficients for the MENBMs with dependent variables of sensitive (0–1 µg/mL ceftiofur) and ≥1, ≥8, ≥16 µg/mL resistant Enterobacteriaceae counts (CFU/g) are shown in Table 3 and Table 4. Model coefficients are on the natural logarithmic scale; hence, they should be interpreted together with other covariables in the model including the intercept. To facilitate interpretation of the MENBMs, non-linear combination estimates for each model’s predicted colony counts are summarized in Table 5. Compared to non-treated cows, there were significantly lower counts of sensitive Enterobacteriaceae from treatment until 36–42 days post-treatment. The point estimates indicate that resistant bacteria, at ≥1 µg/mL ceftiofur, were greater than the sensitive counts until 7–8 days post-treatment. The largest counts for ceftiofur-resistant Enterobacteriaceae, at all ceftiofur concentrations, occurred during the AMD treatment itself. The ≥1 µg/mL ceftiofur model showed significantly increased counts post-treatment until 7 to 8 days, where it briefly returns to pre-treatment levels, before increasing above pre-treatment counts at 15–21 days and again at 36–42 days post-treatment. The ≥8 µg/mL and ≥16 µg/mL models also numerically estimated a similar pattern of cyclical re-emergence of resistance. The ≥8 µg/mL ceftiofur-resistant bacteria required 3 to 4 days post-treatment to return to pre-treatment levels and the ≥16 µg/mL ceftiofur-resistant bacteria were not significantly different from pre-treatment levels by 1–2 days post-treatment.

### 3.4. Multilevel Mixed Effects Interval Regression Models

The “maximum MIC” model estimated higher MIC than the “population MIC” model for all time points (Table 6, Figure 4), as expected. The MENBM models’ treatment variable specification was used here for consistent interpretation. Fixed effects for season, DIM, fecal consistency, lactation and mastitis treatment were all non-significant. For both MIC models, the shapes of their prediction curves were similar, with the “maximum MIC” model peak occurring at 3 to 4 days post-treatment and the “population MIC” model peaking during AMD treatment. The maximum MIC returned to pre-treatment MIC by 5–6 days post-AMD treatment and the population MIC by 7–8 days. A second, smaller peak of MIC was observed at 15–21 days post-AMD treatment in the maximum model but was not observed in the population model.

## 4. Discussion

The current study showed the temporal association between AMD treatment and ceftiofur-resistant Enterobacteriaceae in early lactation dairy cows. In untreated cows, the prevalence of resistant Enterobacteriaceae phenotypes was extremely low, while cows that were treated with systemic AMDs experienced a rapid emergence of AMR phenotypes that peaked in prevalence during treatment (Table 5). The return to pre-treatment prevalence for resistant bacteria varied on the level of ceftiofur resistance being measured, taking approximately 7–8 days for the ≥1 µg/mL ceftiofur resistance counts to return to normal. The resistance dynamics reported here should be considered when designing future AMR surveillance and research projects. Specifically, with regard to the time of sample collection, fecal samples collected from cows in early lactation, where AMD treatment risk is expected to be highest, are likely to have higher fecal resistance profiles compared to other adult dairy cows.

The observed decrease in total Enterobacteriaceae CFU per gram of feces following AMD treatment is consistent with outcomes from previous studies [23,24,25,31]. However, the observed cyclical re-emergence of resistant phenotypes at around 15-day intervals, with diminishing counts at each subsequent peak has not, to the authors’ knowledge, previously been described in adult dairy cows. Such re-emergence of resistance may not have been observed in earlier studies due to short follow-up periods [20,21], long sampling intervals [22,24,25] or too few colonies tested for antimicrobial resistance [21,22]. Observing individual cow data, the phenomenon of re-emergence at 15–21 days post-treatment was observed in 9 out of the 16 treated cows and would, therefore, unlikely be due to unrecorded re-treatment with an antimicrobial. Likewise, husbandry events (e.g., pen changes) are unlikely to explain this observation, as this was not seen in the time-matched untreated cows (Figure 2b–d). We hypothesize that the cyclical re-emergence of resistant phenotypes could have resulted from horizontal gene transfer (HGT) of antimicrobial resistance genes elements within the gut flora, which is known to occur through three main mechanisms: conjugation, transduction and transformation [17,18]. Conjugation and transduction both occur through direct sharing of DNA with donor bacteria and phages, respectively, while transformation is through the uptake of extracellular free DNA [18]. Though the natural competency of *E.coli* for transformation has traditionally been believed to be low [18], this belief has recently come under challenge [17,32]. Following a peak in resistant bacteria, we suggest a relative abundance in extracellular DNA carrying resistant genes would be available for HGT via transformation. Other studies have shown that mice enteric *E.coli* and *Shigella* spp. can readily obtain resistance genes following feeding of extracellular antibiotic resistance genes [33], and *E.coli* is transformation competent in vitro under multiple conditions [32]. Additionally, models of bacteria population dynamics predict oscillating gene abundance when HGT via both conjugation and transformation are considered [34].

A similar longitudinal cohort study investigated the dynamics of calf AMR bacteria by enumerating Enterobacteriaceae on MacConkey agar plates infused with either 0 or 30 µg/mL of ceftiofur [28]. Calves were fed neomycin sulfate infused milk replacer for the first 20 days of life and were treated with ceftiofur for illness events. Counts of resistant bacteria increased during ceftiofur treatment, peaked 3–4 days post-treatment and returned to baseline by 5–6 days post-treatment. The temporal dynamics of ceftiofur resistant bacteria were similar between the calf study and our own. However, the level of resistance was substantially different, with the MENBM analysis at 30 µg/mL in our cohort of adult cows not possible due to few samples with bacterial growth. Further research is needed to understand the difference in the level of resistance between dairy calves and adult lactating dairy cows.

The MIC models reported here offer an important distinction compared to models that determine antimicrobial susceptibility using individual colonies, such as the broth microdilution method [35]. In brief, the microdilution method inoculates media that contain increasing concentrations of an antimicrobial substance with select monocultures of bacteria, each based on a single isolate, to determine the antimicrobial concentration that inhibits that isolate’s growth. The microdilution method is common in commercial labs and research settings but is cost-limiting with regard to the number of colonies that can be analyzed. Limiting the number of isolates reduces the precision of estimates for mean MIC compared to the method described in this paper, which by way of comparison, included over 6.8 × 10^8^ individual colonies in the “population MIC” model. Limitations to the method presented here, compared to broth microdilution, are that specific bacterial colonies are observed at only one AMD concentration and both labor and cost would substantially increase if multiple AMDs were to be tested. Compared to the results reported by Foster et.al. [21], who measured fecal *E.coli* MIC using broth microdilution on eight colonies per sample from steers treated with three days of ceftiofur hydrochloride, our results from the population MIC model had a lower peak MIC (20.45 vs. 8.8 µg/mL) and a longer duration of elevated MIC (3 vs. 7–8 days post-treatment).

Antimicrobial resistance MIC breakpoints for the clinical efficacy of ceftiofur in cattle diseases, besides mastitis, have yet to be established, though a cut-off point of ≥8 µg/mL ceftiofur is often used [27,36]. Our investigation modelled acquired resistance to a fecal commensal bacteria without direct applicability to an enteric disease or specific treatment efficacy. As such, a more appropriate metric for discussing resistance acquisition and its contribution to the resistome may be the Epidemiological Cut-Off Value, ECOFF (EUCAST) [15,37] or, ECV (CLSI) [27]. The ECOFF is a measure that attempts to separate wild-type (WT), non-resistant bacteria from those bacteria that have acquired resistance, often subsequent to an external selective pressure such as AMD exposure. In our study, only 2.72% of bacteria counts from untreated animals had ≥1µg/mL ceftiofur resistance and the population MIC model predicts a pre-treatment mean MIC of 1.09 (SE = 0.016) in the treated cows. Such outcomes are supported by another study on adult California dairy cows that found 4.61% of *E.coli* resistant to ceftiofur at 1 µg/mL [14], and EUCAST reports a WT *E.coli* ECOFF value for ceftiofur at ≤1 µg/mL [38]. These results indicate that WT Enterobacteriaceae have an ECOFF value substantially less than the clinical breakpoint value of ≥8 µg/mL. The relative importance of interpreting study results as either ECOFF or clinical breakpoints values is determined by the study objectives.

A major limitation of our study was the low number of treated animals, with only Farm B treating enrolled cows with systemic AMDs. Using herd-wide treatment frequency and adjusting for lactation group, the post hoc number of enrolled cows that was expected to be treated was 2 cows from Farm A and 19 cows from Farm B. Comparison of periparturient disease frequency is complicated by the two farms adopting substantially different attitudes towards disease monitoring; Farm B hired a dedicated 2-person team to monitor individual fresh cows (until around 20 DIM) twice daily, whilst Farm A had a limited labor force and implemented no formal fresh-cow monitoring system.

It is possible that colony counts were influenced by other measured factors, such as fecal consistency, season or treatment duration, that were not significant in our models. Laboratory procedures may have artificially increased variation in colony counts through a number of pathways that include: sample processing times, homogenization of feces during processing, bacteria surviving freezing/thawing and ceftiofur concentrations in the agar. Variation from these procedures were minimized through quick processing times (~6 h after collection), standardized operating procedures and using sensitive/resistant *E.coli* control strains. It would be expected that these causes of variation would introduce minor non-differential bias. The initial enumeration for both Farm B cohorts and half of Farm A’s first cohort was performed on a Model D spiral plater (Spiral System Inc., Cincinnati, OH, USA). Subsequently we discovered mechanical issues related to the inoculation arm’s start position. As a result, samples with positive colony counts were repeated using a second stored fecal sample on the WASP 2, as described in the methods section, replacing all previous Model D results. We do not expect this alteration to the initial protocol to have introduced any bias.

## 5. Conclusions

The current study shows that counts of ceftiofur-resistant enteric Enterobacteriaceae (CFU/g) from early lactation dairy cows rapidly increased with AMD treatment, and the time to return to pre-treatment levels was determined by level of ceftiofur resistance being measured (≥1 µg/mL ceftiofur-resistant: 7–8 days, ≥8 µg/mL: 3–4 days, ≥16 µg/mL: 1–2 days). Populations of sensitive bacteria (0–1 µg/mL) remained below pre-treatment levels for 36–42 days post-AMD treatment. The MIC models predicted peaks at 3–4 days post-treatment for the maximum model (MIC = 24.7 µg/mL ceftiofur) and during AMD treatment for the population model (MIC = 8.8 µg/mL ceftiofur). Return to pre-treatment levels required 5–6 and 7–8 days for the respective MIC models. Further research is required to validate the findings of this study, including the cyclical re-emergence of resistant phenotypes post-AMD treatment and any contributing role of HGT via transformation.

## Figures and Tables

**Figure 1 microorganisms-09-00828-f001:**
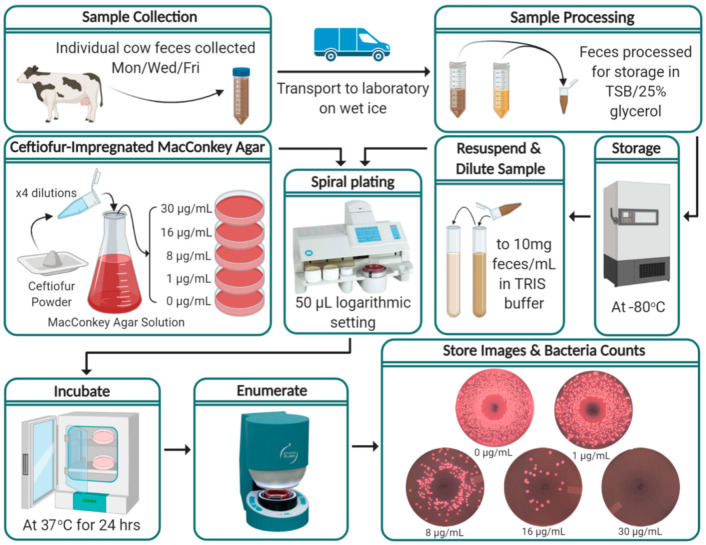
Flow diagram of study protocol from on-farm fecal sample collection through to the enumeration of Enterobacteriaceae and storage of results. (Figure created with BioRender.com; Spiral plating and enumerate steps’ images are credited to Microbiology International, Frederick, MD and IULmicro, Barcelona, respectively).

**Figure 2 microorganisms-09-00828-f002:**
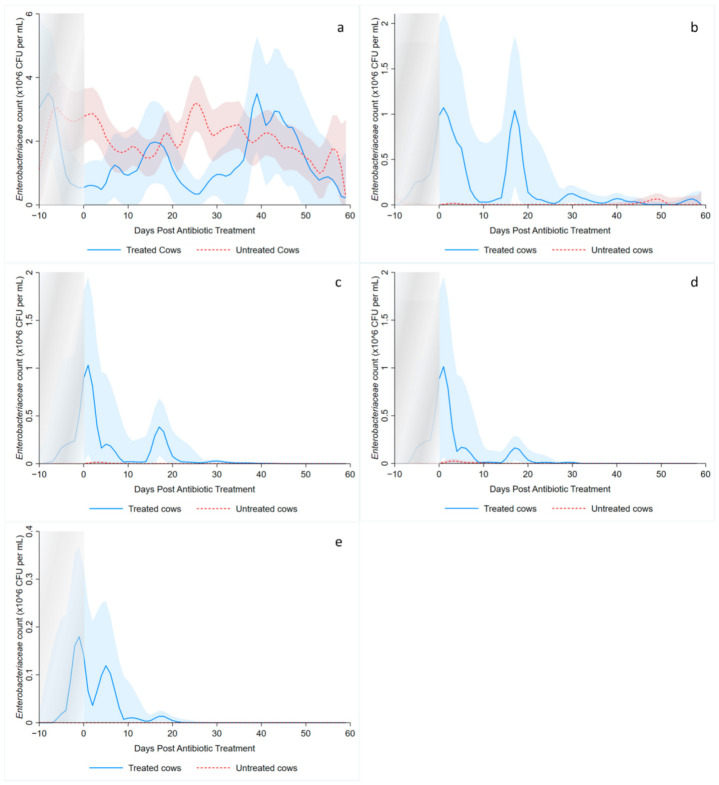
Kernel-weighted local polynomial smoothed regression plots for Enterobacteriaceae counts in cows treated with systemic antimicrobial drugs and time-matched untreated cows by days post-antibiotic treatment. Counts for ceftiofur resistance at (**a**) 0–1 µg/mL (**b**) ≥1 µg/mL (**c**) ≥8 g/mL (**d**) ≥16 µg/mL (**e**) ≥30 µg/mL. Shaded areas around the regression lines are the 95% confidence intervals. For treated cows, data between −10 to 0 days post-antibiotic treatment combine cows currently receiving treatment and pre-treatment cows, due to varying treatment duration (indicated by the shaded box).

**Figure 3 microorganisms-09-00828-f003:**
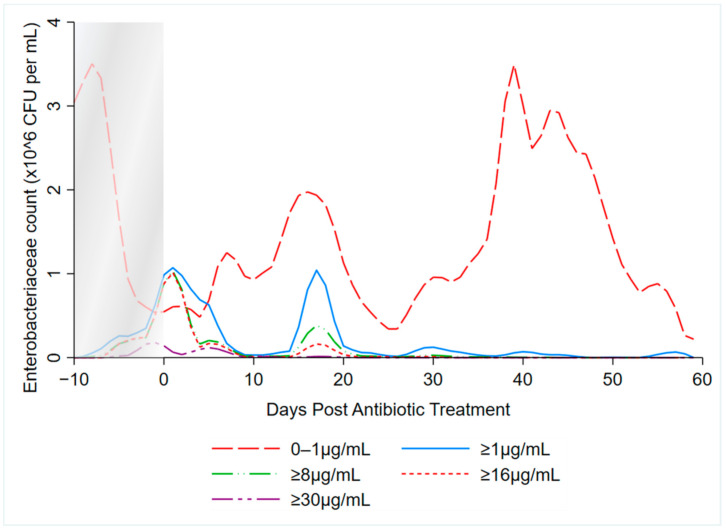
Kernel-weighted local polynomial smoothed regression plots for Enterobacteriaceae counts in cows treated with systemic antimicrobial drugs by days post-antibiotic treatment. Data between −10 to 0 days post-antibiotic treatment combine cows currently receiving treatment and pre-treatment cows, due to varying treatment durations (indicated by the shaded box).

**Figure 4 microorganisms-09-00828-f004:**
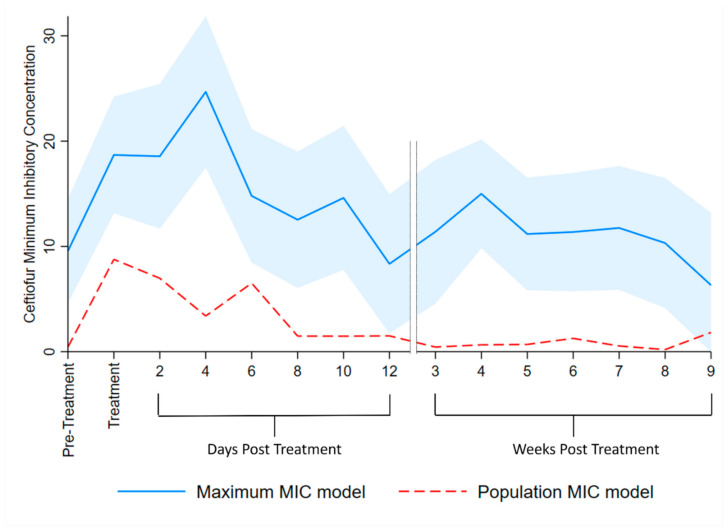
Multilevel mixed effect interval regression analysis for ceftiofur MIC of Enterobacteriaceae estimates. The Maximum MIC model interval data lower limit was determined by the greatest ceftiofur concentration that permitted growth and the upper limit by the lowest ceftiofur concentration that inhibited growth, per cow per sample day. The shaded area represented the 95% confidence interval for the maximum MIC model. The Population MIC model data was determined by estimating the interval-censored total number of colonies/g of feces for each interval of ceftiofur concentration observed, per cow per day. The 95% confidence intervals are censored for the population MIC model as the large count of colonies per cow per day results in too narrow ranges.

**Table 1 microorganisms-09-00828-t001:** Example determination of interval values for the multilevel mixed effects interval regression model for ceftiofur Minimum Inhibitory Concentration (MIC).

Example 1:					
Ceftiofur concentration	0 µg/mL	1 µg/mL	8 µg/mL	16 µg/mL	30 µg/mL
Colony counts ^c^	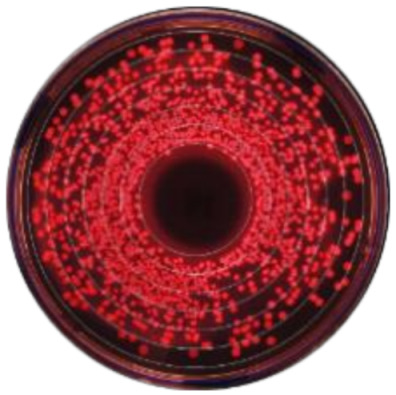 1500	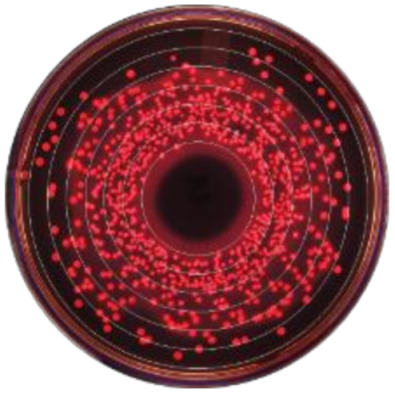 1000	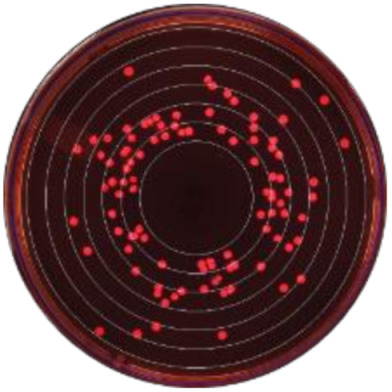 100	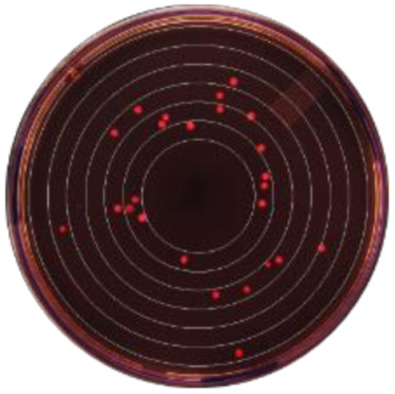 25	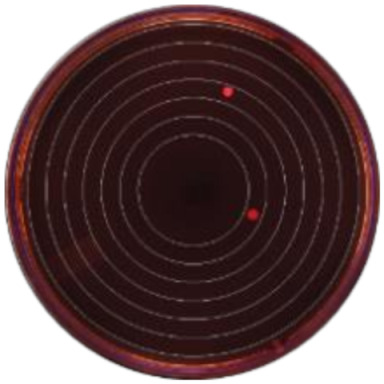 2
Interval specification	[0 to 1]	[1 to 8]	[8 to 16]	[16 to 30]	[30 to +∞)
Maximum MIC ^a^	-	-	-	-	1
Population MIC ^b^	500	900	75	23	2
**Example 2:**					
Ceftiofur concentration	0 µg/mL	1 µg/mL	8 µg/mL	16 µg/mL	30 µg/mL
Colony counts ^c^	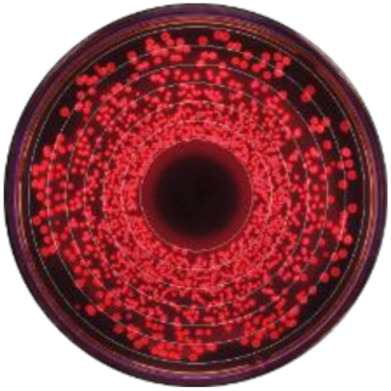 1500	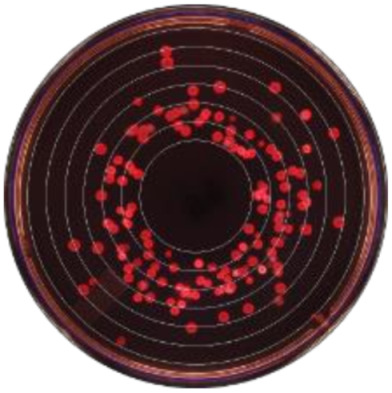 400	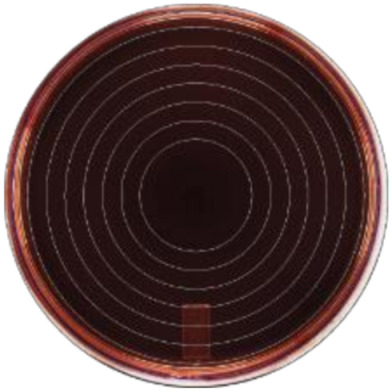 2	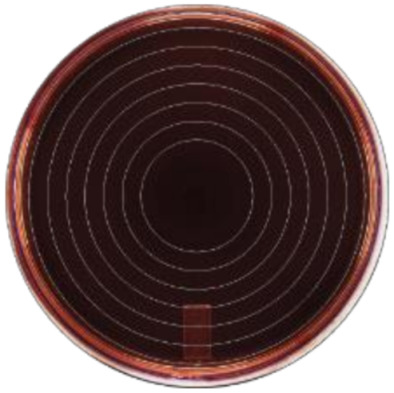 0	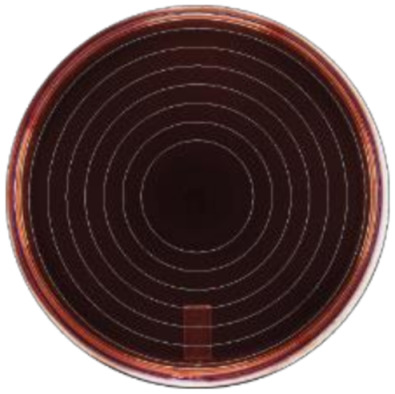 0
Interval specification	[0 to 1]	[1 to 8]	[8 to 16]	[16 to 30]	[30 to +∞)
Maximum MIC ^a^	-	-	1	-	-
Population MIC ^b^	1100	398	2	0	0

^a^ Lower limit of interval determined by highest ceftiofur concentration that permitted growth and upper interval by lowest ceftiofur concentration that inhibited growth. ^b^ Population MIC determined by subtracting the interval’s upper endpoint ceftiofur concentration colony count from the interval’s lower endpoint ceftiofur concentration colony count. The right-censored interval [30, +∞) was the unadjusted 30 µg/mL counts. ^c^ Presented values are for illustrative purposes only.

**Table 2 microorganisms-09-00828-t002:** Enrollment counts, lactation group structure and antimicrobial treatments for study cohorts.

		Study Cows		Antimicrobial Treatments	
	Lactation	Eligible	Enrolled	Injectable	Intramammary
**Farm A Cohorts**					
Winter/Fall					
	1st	30	6	0	0
	2nd	12	6	0	0
	3+	20	12	0	1
Spring/Summer					
	1st	12	12	0	0
	2nd	10	7	0	1
	3+	8	5	0	1
**Farm B Cohorts**					
Winter/Fall					
	1st	39	8	6	0
	2nd	20	6	2	1
	3+	36	10	2	0
Spring/Summer					
	1st	24	8	3	0
	2nd	23	5	0	0
	3+	25	11	3	0

**Table 3 microorganisms-09-00828-t003:** Multilevel mixed effect negative binomial model estimates for “sensitive” (0–1 µg/mL ceftiofur) and ≥1 µg/mL ceftiofur-resistant Enterobacteriaceae counts post-systemic antimicrobial drug treatments in 91 cows on two California dairies over two seasons.

	“Sensitive” Enterobacteriaceae Counts (0–1 µg/mL Ceftiofur)					≥1 µg/mL Ceftiofur-Resistant Enterobacteriaceae Counts				
				95% Confidence Interval					95% Confidence Interval	
*Variable*	Estimate ^a^	S.E.	*p* Value	Lower Limit	Upper Limit	Estimate ^a^	S.E.	*p* Value	Lower Limit	Upper Limit
Pre-treatment	referent					referent				
During treatment	−1.24	0.007	<0.01	−1.26	−1.23	6.00	0.022	<0.01	5.95	6.04
Days post-treatment:										
1–2	−1.53	0.011	<0.01	−1.55	−1.51	4.64	1.023	<0.01	2.63	6.64
3–4	−2.25	0.021	<0.01	−2.29	−2.21	4.23	0.944	<0.01	2.38	6.08
5–6	−2.30	0.013	<0.01	−2.33	−2.28	2.98	0.851	<0.01	1.31	4.64
7–8	−1.45	0.000	<0.01	−1.45	−1.45	0.64	0.749	0.40	−0.83	2.10
9–10	−1.71	0.002	<0.01	−1.71	−1.70	0.99	0.093	<0.01	0.81	1.17
11–12	−0.48	0.069	<0.01	−0.62	−0.34	0.49	0.271	0.07	−0.04	1.02
13–14	−1.02	0.004	<0.01	−1.03	−1.02	1.06	0.977	0.28	−0.86	2.98
15–21	−0.62	0.021	<0.01	−0.66	−0.57	2.93	1.253	0.02	0.47	5.39
22–28	−1.28	0.042	<0.01	−1.36	−1.20	0.27	0.419	0.52	−0.55	1.09
29–35	−0.61	0.047	<0.01	−0.70	−0.52	1.24	0.704	0.08	−0.14	2.62
36–42	0.32	0.040	<0.01	0.24	0.40	1.23	0.263	<0.01	0.71	1.74
43–49	0.51	0.063	<0.01	0.39	0.64	−0.68	0.597	0.26	−1.85	0.49
50–56	−0.34	0.041	<0.01	−0.42	−0.26	−1.05	1.387	0.45	−3.77	1.66
57–58	−1.99	0.010	<0.01	−2.01	−1.97	−0.56	0.151	<0.01	−0.85	−0.26
Farm										
A	referent					referent				
B	0.09	0.042	0.034	0.01	0.17	2.32	0.716	<0.01	0.92	3.72
Intercept	14.15	0.023	<0.01	14.10	14.19	6.77	0.804	<0.01	5.19	8.34

Ln(dispersion)	0.65	0.260		0.14	1.16	3.47	0.511		2.47	4.47

*Random effects*										
DIM ^b^ (slope)	<0.01	<0.01		<0.01	<0.01	<0.01	0.004		<0.01	0.45
Cow (intercept)	1.12	0.389		0.56	2.21	2.05	2.256		0.24	17.70

^a^ Negative binomial model coefficient comparing the explanatory variable’s identified level to the respective referent. Coefficients are magnitudes of change on the natural logarithm scale and hence should be interpreted with the intercept and any other covariate, ^b^ Days In Milk.

**Table 4 microorganisms-09-00828-t004:** Multilevel mixed effect negative binomial estimates for ≥8 µg/mL and ≥16 µg/mL ceftiofur-resistant Enterobacteriaceae counts post-systemic antimicrobial drug treatments in 45 cows on one California dairy (Farm B) over two seasons.

	≥8 µg/mL Ceftiofur-Resistant Enterobacteriaceae Counts					≥16 µg/mL Ceftiofur-Resistant Enterobacteriaceae Counts				
				95% Confidence Interval					95% Confidence Interval	
*Variable*	Estimate ^a^	S.E.	*p* Value	Lower Limit	Upper Limit	Estimate ^a^	S.E.	*p* Value	Lower Limit	Upper Limit
Pre-treatment	referent					referent				
During treatment	5.53	1.648	<0.01	0.00	2.30	6.57	1.947	<0.01	2.75	10.38
Days post-treatment:										
1–2	4.47	2.052	0.03	0.03	0.44	4.98	2.761	0.07	−0.43	10.39
3–4	3.02	1.907	0.11	0.11	−0.72	2.83	2.691	0.29	−2.45	8.10
5–6	2.60	1.812	0.15	0.15	−0.95	2.32	2.576	0.37	−2.73	7.37
7–8	−1.06	1.739	0.54	0.54	−4.46	−1.54	2.442	0.53	−6.33	3.24
9–10	0.75	1.915	0.70	0.70	−3.01	−2.07	2.873	0.47	−7.70	3.57
11–12	−0.18	2.003	0.93	0.93	−4.10	−2.95	3.834	0.44	−10.47	4.56
13–14	−0.26	2.084	0.90	0.90	−4.35	−4.06	4.106	0.32	−12.11	3.99
15–21	0.85	1.457	0.56	0.56	−2.00	2.14	1.957	0.28	−1.70	5.97
22–28	−1.43	1.520	0.35	0.35	−4.41	−0.32	2.275	0.89	−4.77	4.14
29–35	−1.57	1.543	0.31	0.31	−4.60	−3.00	2.001	0.13	−6.92	0.93
36–42	−1.47	1.530	0.34	0.34	−4.47	−6.75	2.353	<0.01	−11.36	−2.14
43–49	−3.47	1.718	0.04	0.04	−6.83	−4.81	2.941	0.10	−10.57	0.95
Intercept	9.18	0.553	<0.01	0.00	8.10	8.34	0.709	<0.01	6.95	9.73

Ln(dispersion)	3.31	0.069		3.18	3.45	4.03	0.096		3.84	4.22

*Random effects*										
DIM ^b^ (slope)	0.03	0.012		0.01	0.07	0.03	0.010		0.01	0.06
Cow (intercept)	4.25	1.398		2.23	8.10	4.70	1.941		2.10	10.56

^a^ Negative binomial model coefficient comparing the explanatory variable’s identified level to the respective referent. Coefficients are magnitudes of change on the natural logarithm scale and hence should be interpreted with the intercept. ^b^ Days In Milk.

**Table 5 microorganisms-09-00828-t005:** Prediction estimates based on multilevel mixed effect negative binomial models of Enterobacteriaceae counts post-systemic antimicrobial drug treatment at four different levels of ceftiofur resistance. Bolded estimates were calculated from statistically significant coefficients (*p* < 0.05) in their respective models, bracketed values are the 95% confidence intervals.

	Enterobacteriaceae (x,1000) per Gram of Feces at Respective Levels of Ceftiofur							
Time	<1 µg/mL (Sensitive) ^a^		≥1 µg/mL ^a^		≥8 µg/mL		≥16 µg/mL	
Pre-treatment/Untreated	1526	(1469–1584)	9	(7–10)	10	(0–20)	4	(0–10)
During Treatment	**440**	(417–463)	**3570**	(2802–4338)	**2437**	(0–9853)	**2979**	(0–14,000)
Days Post-Treatment:								
1–2	**331**	(311–351)	**917**	(0–2914)	**845**	(0–4166)	611	(0–3823)
3–4	**161**	(160–162)	**612**	(0–1850)	198	(0–934)	71	(0–438)
5–6	**153**	(151–155)	**174**	(0–494)	131	(0–585)	43	(0–248)
7–8	**359**	(346–372)	17	(0–44)	3	(0–14)	1	(0–5)
9–10	**277**	(265–288)	**24**	(15–32)	21	(0–95)	1	(0–3)
11–12	**945**	(852–1037)	14	(4–25)	8	(0–39)	0	(0–2)
13–14	**548**	(531–565)	26	(0–79)	7	(0–37)	0	(0–1)
15–21	**825**	(822–827)	**166**	(0–603)	23	(0–82)	35	(0–160)
22–28	**424**	(405–442)	12	(0–23)	2	(0–9)	3	(0–16)
29–35	**831**	(785–877)	31	(0–78)	2	(0–8)	0	(0–1)
36–42	**2110**	(2023–2196)	**30**	(9–51)	2	(0–8)	**0**	(0–0)
43–49	**2552**	(2335–2768)	5	(0–11)	**0**	(0–1)	0	(0–0)
50–56	**1090**	(1044–1136)	3	(0–12)	– ^b^		– ^b^	
57–58	**209**	(205–212)	**5**	(3–7)	– ^b^		– ^b^	

^a^ Estimates derived for Farm B only. ^b^ Negative binomial model coefficients were not specified for these values of days post-treatment.

**Table 6 microorganisms-09-00828-t006:** Multilevel mixed effect interval regression output for the Maximum ^a^ and Population ^b^ ceftiofur MIC ^c^ models for Enterobacteriaceae, post-systemic antimicrobial drug treatments in 16 cows on a California dairy over two seasons.

	Maximum MIC					Population MIC				
				95% Confidence Interval					95% Confidence Interval	
*Variable*	Estimate ^d^	S.E.	*p* Value	Lower Limit	Upper Limit	Estimate ^d^	S.E.	*p* Value	Lower Limit	Upper Limit
Pre-treatment	referent					referent				
During treatment	9.13	2.541	<0.01	4.15	14.11	8.33	0.0003	<0.01	8.329	8.330
Days post-treatment:										
1–2	9.00	3.232	0.01	2.67	15.34	6.53	0.0005	<0.01	6.534	6.536
3–4	15.12	3.392	<0.01	8.47	21.77	2.95	0.0009	<0.01	2.949	2.953
5–6	5.24	2.925	0.07	−0.49	10.97	6.07	0.0011	<0.01	6.068	6.073
7–8	2.98	3.006	0.32	−2.91	8.87	1.04	0.0004	<0.01	1.044	1.046
9–10	5.05	3.206	0.12	−1.24	11.33	1.04	0.0010	<0.01	1.034	1.038
11–12	−1.21	3.070	0.69	−7.23	4.80	1.06	0.0010	<0.01	1.059	1.063
13–14	1.83	3.209	0.57	−4.46	8.12	0.00	0.0007	0.51	−0.002	0.001
15–21	5.44	2.258	0.02	1.01	9.86	0.21	0.0003	<0.01	0.210	0.211
22–28	1.62	2.371	0.49	−3.03	6.27	0.25	0.0008	<0.01	0.251	0.254
29–35	1.81	2.525	0.47	−3.14	6.76	0.83	0.0005	<0.01	0.825	0.827
36–42	2.19	2.677	0.41	−3.05	7.44	0.10	0.0004	<0.01	0.100	0.101
43–49	0.76	2.840	0.79	−4.80	6.33	−0.24	0.0004	<0.01	−0.241	−0.240
50–56	−3.24	3.229	0.32	−9.57	3.09	1.39	0.0007	<0.01	1.389	1.392
Intercept	9.56	2.510	<0.01	4.64	14.48	0.44	0.0002	<0.01	0.439	0.440

*Random effects*										
DIM ^e^ (slope)	0.02	0.011		<0.01	0.06	0.01	<0.0001		0.009	0.009
Cow (intercept)	68.20	29.254		29.42	158.09	1.38	0.0004		2.287	2.288
Residual variance	82.34	7.328		69.16	98.03	10.97	0.0004		10.971	10.972

^a^ Lower limit of interval determined by highest ceftiofur concentration that permitted growth and upper interval by lowest ceftiofur concentration that inhibited growth, per cow, per sample day. ^b^ Interval-censored total Enterobacteriaceae count/g of feces for each interval was estimated per cow, per sample day. ^c^ Minimum Inhibitory Concentration. ^d^ Interval regression model coefficient comparing the explanatory variable’s identified level to the respective referent. ^e^ Days In Milk.

## Data Availability

This study was sponsored by the California Department of Food and Agriculture and is subject to California Food and Agriculture Code (FAC) Sections 14400 to 14408. FAC section 14407 requires that data collected be held confidential to prevent individual identification of a farm or business; as such, raw data from this study is not able to be shared.

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
