# Peer review of "Effect of Antimicrobial Treatment on the Dynamics of Ceftiofur Resistance in Enterobacteriaceae from Adult California Dairy Cows"

_microorganisms, 2021, doi:10.3390/microorganisms9040828_

Round 1
Reviewer 1 Report
Sheedy and colleagues showed the Effect of antimicrobial treatment on the dynamics of ceftiofur resistance in Enterobacteriaceae from adult California dairy cows. They tested two dairies including 96 cows with their fecal samples and they counting the bacterial count of the family Enterobactericease by plating on MacConkey. The author showed that no bacteria were grown in 88% of samples collected from non-AMD treated cows at 8µg/mL ceftiofur, while samples from AMD treated cows had peak counts of resistant Enterobacteriaceae during AMD treatment and returned to baseline counts by 3-4 days post treatment at 8µg/mL. The authors concluded that the effect of systemic ceftiofur on the resistance of Enterobacteriaceae in early lactation dairy cows was limited.
The manuscript is well written. I have the following questions
1- The authors need to characterize which Enterbactericeae isolates present in the cow feces.
2-The authors need to show genetic resistance to ceftiofur such as the blaCMY-2-positive isolates
3- The authors include some references such as ref 28 (which is not published paper, In preparation). This should be corrected.
Author Response
Reviewer 1:
I have the following questions
Reviewer 1: The authors need to characterize which Enterbactericeae isolates present in the cow feces.
Author: It was not feasible to characterize the Enterobacteriaceae beyond the family level due to the large number of isolates counted per sample; often 200-400 colonies were counted per 0µg/mL ceftiofur plate, of which a little over 4,500 plain MacConkey plates were counted. Lines 177-180 read “Enterobacteriaceae enumeration was performed using the SphereFlash Automatic Colony Counter (IULmicro, Barcelona), counting only the lactose-fermenting ‘pink colonies’ and with manual edits to correct misclassified colonies or artefacts, as necessary.” to define what constituted as count in our analysis. After this sentence we have added “Further characterization of colonies was not performed” (Line 181).
Reviewer 1: The authors need to show genetic resistance to ceftiofur such as the blaCMY-2-positive isolates
Author: We appreciate this suggestion, and we have a keen interest in investigating the underlying genetic determinants that explain the observed trend and population waves of resistant bacteria following antimicrobial treatment. Since our study analyzed phenotypic resistance at the population level, we think metagenomic analyses of the fecal samples, rather than detecting the presence of specific genes on selected single colonies, would be the most suitable approach to generate a parallel population level data on the changes in total microbial populations and associated antimicrobial resistance determinants, that will allow rational comparison with the phenotypic data. While metagenomic analysis was outside the scope of the current study, which is focused on population level phenotypic resistance, our follow-up studies will aim at generating this additional information to further our understanding of the impact of (ceftiofur) treatment on the population dynamics of the gut bacteria. We amended the objective statement in the manuscript (Line 81) to underscore the focus of the current study on population phenotypic resistance.
Reviewer 1: The authors include some references such as ref 28 (which is not published paper, In preparation). This should be corrected.
Author: We have corrected the in-text references to remove the terms ‘IN PRESS’ and “IN PREPARATION” (Line 56, 167, 499) and adjusted the reference list as per the available online Instructions for authors for unpublished data (Lines 628-631 and 668-670).

Reviewer 2 Report
In the manuscript ID: microorganisms-1165031 by Sheedy and colleagues, the authors describe a prospective study on the dynamics of antimicrobial resistance in dairy farms in cows, treated with systemic ceftiofur, during the first 60 days of the lactation period. Drug-resistant bacterial counts showed a peak in the first days after antibiotic treatment and after 18 days, exhibiting similar pathways, though at different magnitudes, when bacteria were exposed to different antibiotic concentration. The authors concluded that antibiotic treatment exerted a time-limited action on the development of drug resistance, although future studies are needed to confirm and to characterize in deep the mechanisms undergoing antimicrobial resistance emergence.
The manuscript is clear and well written, the methods and results are scrupulously detailed, well discussed and supported by statistical analysis. The topic is of great interest, it fits perfectly with the aim of the proposed special issue and it arises curiosity since it focuses on the resistance development dynamics, rather than on its detection.
Have the authors considered sequencing bacterial strains recovered over time in order to elucidate the drug resistance determinants? It would be interesting to analyse whether ceftiofur resistance is mainly due to mutations or additional resistance genes and, in this second case, if such genes can be effectively transferred to the gut flora.
As minor revision, please move the text corresponding to lines 174-186 to discussion, as the “Material and methods” section should include only the “crude” protocols; any explanation should be addressed in the discussion section.
After these minor revisions, the manuscript can be published in “Microorganisms”.
MINOR COMMENTS
Lines 35, 62, 73 and 473, as “AMR” stands for “antimicrobial resistance”, please correct with “drug resistant bacteria”;
Lines 55, 57 and 141, please correct “Gram-negative”;
Line 224, please correct “due to the failure of model convergence”;
Line 366, please correct “statistical significance of 14 days post treatment”;
Line 470, please type “in vitro” in Italic;
Author Response
Reviewer 2:
The manuscript is clear and well written, the methods and results are scrupulously detailed, well discussed and supported by statistical analysis. The topic is of great interest, it fits perfectly with the aim of the proposed special issue and it arises curiosity since it focuses on the resistance development dynamics, rather than on its detection.
Reviewer 2: Have the authors considered sequencing bacterial strains recovered over time in order to elucidate the drug resistance determinants? It would be interesting to analyse whether ceftiofur resistance is mainly due to mutations or additional resistance genes and, in this second case, if such genes can be effectively transferred to the gut flora.
Author: We whole-heartedly agree with this sentiment. Our initial funding from the CDFA was explicitly conditioned on investigating the phenotypic expression of resistance and were not interesting in funding genetic exploration when approached following the results of this study. We are currently looking for funding opportunities to support investigation into the genetic profiles of these isolates.
Reviewer 2: As minor revision, please move the text corresponding to lines 174-186 to discussion, as the “Material and methods” section should include only the “crude” protocols; any explanation should be addressed in the discussion section.
Author: We believe it is important for the reader to know why 16µg/mL ceftiofur was excluded from Farm A during the statistical analysis methods section and respectively disagree on its exclusion from the Material and Methods.
The topic of the Model D spiral plater malfunction has been moved to the end of the discussion session where similar topics of study limitations are being discussed (Line 561 – 568)
Reviewer 2: Lines 35, 62, 73 and 473, as “AMR” stands for “antimicrobial resistance”, please correct with “drug resistant bacteria”;
Author: We would prefer to use the term antimicrobial resistant bacteria over drug resistant bacteria as we feel that ‘drug’ is too broad a term when the study is focused primarily on ceftiofur. We acknowledge that “Antimicrobial Resistance Bacteria” is grammatically incorrect and have adjusted the first instance that we used AMR bacteria to correct this, as below (Line 36):
The One Health concept and antimicrobial stewardship guidelines necessitate that the food animal industry investigate on-farm antimicrobial drug (AMD) use as a risk factor for the selection and emergence of antimicrobial resistant bacteria [3], referred to hereafter as AMR bacteria.
Reviewer 2: Lines 55, 57 and 141, please correct “Gram-negative”;
Author: Corrected
Reviewer 2: Line 224, please correct “due to the failure of model convergence”;
Author: Corrected
Reviewer 2: Line 366, please correct “statistical significance of 14 days post treatment”;
Author: Corrected to read: “The change from two-day to 7-day intervals was based on the loss of statistical significance at 14 days post treatment in the 1µg/mL ceftiofur-resistant bacteria counts.” Line 382
Reviewer 2: Line 470, please type “in vitro” in Italic;
Author: Corrected
Line 308: Table 1 has been edited following feedback from colleagues after presenting the research results at a symposium. The use of visual aides to explain the specification of the interval data was reported to be helpful and Table 1 has been edited as such.

Round 2
Reviewer 1 Report
I read the authors' replies to my concerns. Although the authors did not provide extra-experiments in the revised manuscript, I can understand the challenges. Hope the authors consider these suggestions for future research. In general, the manuscript is good and should be published.